# Ionic Polymer Nanocomposites Subjected to Uniaxial Extension: A Nonequilibrium Molecular Dynamics Study

**DOI:** 10.3390/polym13224001

**Published:** 2021-11-19

**Authors:** Ahmad Moghimikheirabadi, Argyrios V. Karatrantos, Martin Kröger

**Affiliations:** 1Polymer Physics, Department of Materials, ETH Zurich, Leopold-Ruzicka-Weg 4, CH-8093 Zurich, Switzerland; 2Materials Research and Technology, Luxembourg Institute of Science and Technology, 5, Avenue des Hauts-Fourneaux, L-4362 Esch-sur-Alzette, Luxembourg

**Keywords:** strain hardening, elongation, self-healing, mechano-ionic switch, solid polymer electrolyte

## Abstract

We explore the behavior of coarse-grained ionic polymer nanocomposites (IPNCs) under uniaxial extension up to 800% strain by means of nonequilibrium molecular dynamics simulations. We observe a simultaneous increase of stiffness and toughness of the IPNCs upon increasing the engineering strain rate, in agreement with experimental observations. We reveal that the excellent toughness of the IPNCs originates from the electrostatic interaction between polymers and nanoparticles, and that it is not due to the mobility of the nanoparticles or the presence of polymer–polymer entanglements. During the extension, and depending on the nanoparticle volume fraction, polymer–nanoparticle ionic crosslinks are suppressed with the increase of strain rate and electrostatic strength, while the mean pore radius increases with strain rate and is altered by the nanoparticle volume fraction and electrostatic strength. At relatively low strain rates, IPNCs containing an entangled matrix exhibit self-strengthening behavior. We provide microscopic insight into the structural, conformational properties and crosslinks of IPNCs, also referred to as polymer nanocomposite electrolytes, accompanying their unusual mechanical behavior.

## 1. Introduction

Recently, a new family of nanocomposites by Odent et al. [1], Potaufeux et al. [2], based on ionically functionalized nanosilicas mixed with oppositely charged imidazolium (cationic)-functionalized polyurethanes (PUs), presented impressive mechanical properties. These systems exhibited an attractive electrostatic interaction [3,4,5,6,7,8] between nanosilicas and the PU matrix. In particular, two main interesting features have been observed in ionic PU/nanosilicas composites [1]: (a) upon increasing the volume fraction of nanosilicas (up to 20 wt%), an increase of strain at break was observed; (b) by increasing the strain rate a simultaneous increase of stiffness and toughness was detected [1,9]. While the increase of stiffness (Young modulus) can be interpreted by the addition of nanosilicas into the PU matrix; however, the mechanisms that lead to the improvement of strain at break and toughness in nanocomposites remain open questions [10]. According to Odent et al. [1], both improvements do not originate from the nature of the PU matrix, but rather the nanoparticles’ (NPs) mobility during deformation, while ionic bonds break and reform with polymer chains as NPs move. In the work of Shah et al. [10] on poly(vinylidene fluoride) and polystyrene (PS) nanocomposites (containing 5 wt% of nanoclay), the mobility of the NPs was dictated by the matrix mobility, and an energy dissipation mechanism was introduced [11] that led to an enhanced toughness via the orientation and alignment of chains under stress. Specifically, NPs could retard the growth of cracks, leading to an improved toughness [10].

There have been a few molecular simulation efforts using either atomistic or coarse-grained models [12,13,14,15,16,17] focusing on the mechanical behavior of traditional (nonionic) polymer nanocomposites. In particular, in a coarse-grained molecular dynamics (MD) study, the toughness of a nanocomposite was shown to be correlated with the minimization of nanofiller diffusion compared to the polymer matrix [18]. This result is in contrast to the work of Shah et al. [10] and the coarse-grained simulation work by Gersappe [11]. In the latter, it was demonstrated that the mobility of nanofillers improves the toughness of the nanocomposite through energy dissipation [11]. Riggleman et al. [19] showed that the stress–strain curve of traditional (nonionic) nanocomposites exhibits an initial elastic behavior followed by yielding and then a strain softening when the uniaxial tensile strain reached ≈ 5%. For strains beyond 10%, a strain hardening [20] (stress increase with strain beyond the yield point) was reported to set in. In another study with high NP volume fractions, polymer bridging between the NPs resulted in an enhancement of the mechanical properties [21]. At even larger volume fractions, a strain softening of the nanocomposite beyond the yield point was observed by Lin et al. [22]. In addition, Hagita et al. [23], using the same molecular model adapted for crosslinked polymers, reported that the stress–strain behavior of the nanocomposite is influenced by the type of interaction between polymers and NPs. For their small mesh cross-linked polymer networks, polymer–NP interactions had a small effect on NP configurations during uniaxial extension while starting from the same NP morphology [24]. The mechanical properties could even monotonically improve with the increase of polymer–NP interaction strength due to the tele-bridge structure formations between polymers and NPs [25]. The shear and Young’s moduli were enhanced in comparison with those in a polymer matrix for attractive and neutral NPs [26]. The NP volume fraction had an effect on the stress–strain behavior of the nanocomposite due to the network formed between polymers and nanofillers; in particular, the stress increased with NP volume fraction in the linear elastic regime (at 10% strain), and the higher surface-to-volume ratio of the NPs was attributed to contribute to the mechanical improvement of the nanocomposite [27]. Moreover, the grafting of NPs with polymers was reported to also reinforce the nanocomposite and increase the elastic constants [28,29,30]. It was also found that strain hardening was only weakly affected by grafting density or NP size [31].

Atomistic simulations are more time-consuming, rely on dedicated force models and usually consider only a very few NPs. In particular, Frankland [32] performed atomistic MD on polyethylene–carbon nanotube (PE-CNT) nanocomposites and evaluated the stress–strain curves for different architectures of single wall CNTs in the PE matrix, when the strain was applied parallel and transverse to the CNT alignments. It was suggested that the use of higher CNT aspect ratio can improve the stress–strain behavior [33]. Moreover, NP surface functionalization resulted in stronger interfacial interactions with the polymer matrix and thus in an improvement of mechanical properties [34]. Another atomistic effort on a nanosilica/polyimide mixture showed an increase of Young’s and shear moduli in comparison to the polyimide matrix [35]. The addition of nanosilicas into a PS matrix resulted in an enhanced Young’s modulus [36]. The same trend was observed for sulfur-crosslinked cis-1,4-polybutadiene with nanosilicas [37]. For attractive nanocomposites, the failure stress of the nanocomposite was correlated to NP loading, surface area of the formed interphase, and the NP network [38]. In such a case, a glassy polymer was formed around the NPs [39]. It was shown further that both attractive and neutral NPs increased the Young’s modulus in comparison to the unfilled polymer matrix, whereas repulsive NPs were found to decrease the moduli [39]. Moreover, the elastic moduli of the nanocomposite varied with the NP diameter at a fixed NP loading [38].

In this work, we study the behavior of ionic polymer nanocomposites (IPNCs) subjected to a uniaxial extension at various strain rates, using 8=2×2×2 qualitatively different systems. This allows us to explore the effects of NP content (small and large volume fractions), polymer chain length (entangled versus basically unentangled), and the strength of the ionic interactions (small and large dielectric constant [40]), c.f. Table 1. While experimental [1,41] and equilibrium simulation [8,42] studies are available for some of the systems, to the best of our knowledge, IPNCs have not been studied by nonequilibrium computer simulations so far. We are going to reveal the role of ionic interactions on the enhanced mechanical behavior observed in IPNCs, in comparison with traditional nanocomposites.

The manuscript is organized as follows. In Section 2, we describe the models and simulation method used in this study. As part of the results and discussion Section 3, we investigate the stress–strain behavior in Section 3.1. Subsequently, the polymer and NP structure during the extension are reported in Section 3.2 and Section 3.3. Monitoring the entanglement network (Section 3.4) and the formation of temporary ionic crosslinks (Section 3.5) are shown to help explain the molecular mechanism accompanying the deformation process in IPNCs. The picture is completed with a pore size analysis in Section 3.6, and a detailed investigation of self-strengthening behavior at low strain rates in Section 3.7. Finally, conclusions of this study are drawn in Section 4.

## 2. Methods


### 2.1. Model Description and Setup

In this work, we have performed coarse-grained nonequlibrium molecular dynamics simulations of model IPNCs. While this model has been described thoroughly in our previous work on the equilibrium properties of IPNCs [42], we discuss its essential features in the following for the sake of completeness.

Our model systems are composed of spherical NPs and multibead-spring linear polymer chains (so-called Kremer–Grest polymers [44,45,46]), where each bead represents a number of monomers [47,48]. Adjacent beads within chains are connected by anharmonic springs, while the impenetrable NPs are modeled as rigid, mobile objects whose surfaces are covered by NP surface beads. Polymeric and surface beads may or may not carry a permanent charge, while the systems are overall neutral in each case, to satisfy the charge balance between ionic polymers and nanoparticles [1]. For both NP volume fractions investigated, the ionic polymers keep their chemistry, that is, their unaltered charge density, as in the experiment [1]. We do not include counterions to compensate for the charge, since they do not appear in the experimental nanocomposite system [1] as well. This implies that the NP charge depends on the NP volume fraction and that the effects of charge and NP volume fraction cannot be explored independently, without altering the polymer dimensions. The existence of charges on the polymers and surface beads within our nanocomposite model is the only factor that leads to NP dispersion in comparison with previous simulation models for traditional nanocomposites [17,19,21,24], in which a short range attractive interaction (nonionic) between NPs and monomers promotes dispersion.

We use Lennard–Jones (LJ) units throughout this manuscript, so that all quantities are given as non-dimensional numbers. They receive their dimensional counterparts through their physical units and the dimensional length, time, mass, and charge units, which we (can) assign afterwards [49], so that all results presented potentially hold up for arbitrary choices of real materials.

Specifically, adjacent polymer beads *i* and *j* separated by spatial distance $r_{ij}$ within polymer chains, each made of *N* beads, are connected using finitely extendable nonlinear elastic (FENE) springs [50,51,52,53,54,55,56]:(1)$$V_{ij}^{\mathrm{FENE}}=-\frac{1}{2}kR_{0}^{2}\ln\left(1-\frac{r_{ij}^{2}}{R_{0}^{2}}\right),$$
where in applying Equation (Equation 1), the maximum bond length and spring coefficient are set to $R_{0}=1.5$ and k=30, respectively, as in previous works on neutral polymers [51]. All the polymer and NP surface beads interact via a truncated, purely repulsive LJ potential $V_{ij}^{\mathrm{LJ}}$, whose corresponding force acts along the line connecting the centers of mass of two beads [57]. It is given by:(2)$$V_{ij}^{\mathrm{LJ}}=4\left(\frac{\sigma_{ij}^{12}}{r_{ij}^{12}}-\frac{\sigma_{ij}^{6}}{r_{ij}^{6}}\right),\;r_{ij}\leq 2^{1/6}\sigma_{ij},$$
where $r_{ij}$ represents the spatial distance between any beads i≠j. The polymer entanglement length [58] of this Kremer–Grest model of a polymer melt in the absence of NPs is $N_{e}\approx 86$ by the S-coil estimator [59,60], the bond length is approximately unity, and the characteristic ratio is $C_{\infty }\approx 1.7$ at our chosen temperature T=1.25. The Lorentz–Berthelot mixing rule [57] $\sigma_{ij}=\left(\sigma_{i}+\sigma_{j}\right)/2$ is used for each pair of beads; $\sigma_{i}=1$ if bead *i* belongs to the set of polymer beads, and $\sigma_{i}=0.4$ if *i* belongs to the set of NP surface beads. In addition, the coulombic interaction between charged beads in the polymer matrix (with a relative dielectric constant of $\varepsilon_{r}$) and on the NP surface is incorporated and given by:(3)$$V_{ij}^{\mathrm{Coulomb}}=\frac{q_{i}q_{j}}{\varepsilon_{r}r_{ij}},$$
with $q_{i}=+q$ for a charged polymer bead and $q_{i}=-q$ for a charged NP surface bead, where *q* is the elementary charge of a proton *e* in LJ units, that is, $q=e/\sqrt{4\pi \varepsilon_{0}\sigma \epsilon }$. All NPs are equally and negatively charged; *Q* of its surface beads carry a negative charge −q. The molecular dynamics simulations were performed using the LAMMPS package [61]. The long-range electrostatics were computed using the particle–particle particle–mesh (PPPM) method [62] with an accuracy of $10^{-4}$. The modeled IPNCs consist of monodisperse spherical rigid NPs, each with a baseline radius of 3.75 (implying an effective NP radius of $r_{\mathrm{NP}}=3.75+0.7/2=4.1$), and fully covered by 720 surface beads. The mass of an NP surface bead, $m_{\mathrm{NP}}=0.49$, is chosen so that the NP mass density, calculated as $\rho_{\mathrm{NP}}=720\times m_{\mathrm{NP}}/V_{\mathrm{NP}}$ with NP volume $V_{\mathrm{NP}}=4/3\pi r_{\mathrm{NP}}^{3}$, is ≈1.5 times the mass density of the polymer matrix with monomer mass of m=1, calculated from ρ=nNm/V(1−ϕ) with the simulation box volume *V*, and NP volume fraction $\phi =n_{\mathrm{NP}}V_{\mathrm{NP}}/V$. The start configurations of the simulation cells were created by replicating already equilibrated nanocomposite structures [56]. The large simulation cells were further equilibrated for the duration of approximately five times the relaxation time of the gyration radius. All results to be presented are averages over ten independently generated nanocomposite systems.

### 2.2. Choice of Systems

In the following, we estimate and then set the Bjerrum length, $\lambda_{B}$, through an (approximate) mapping of the simulated systems with existing materials. Imidazolium-functionalized polyurethane (PU)/silica [1] and poly(ethylene oxide) (PEO)/silica [63] nanocomposites have been studied in the melt state at around T≈333 K, thus implying $\epsilon \approx 266\;k_{B}$K for the LJ energy unit, and the nanosilica diameter of around 10 nm that results in σ≈1 nm for the LJ length unit. Therefore, the parameter *q*, which is the ratio between the elementary charge *e* and LJ charge unit $\sqrt{4\pi \epsilon_{0}\sigma \epsilon }$, is calculated and set as q≈7.92. The Bjerrum length $\lambda_{B}$ indicates the length scale at which the magnitude of the electrostatic interaction equals the thermal energy. It is thus obtained from Equation (Equation 3) as:(4)$$\lambda_{B}=\frac{q^{2}}{\varepsilon_{r}T}.$$

While depending on molecular weight and the temperature the dielectric constant of PEO melt varies, as discussed by Koizumi and Hanai [64], Porter and Boyd [65], we consider a dielectric constant of $\varepsilon_{r}=24$, which is around the values reported [64,65], implying a Bjerrum length of $\lambda_{B}\approx 2.1$. We also study the effect of ionic strength by simulating a set of replicate systems with a different dielectric constant of $\varepsilon_{r}=50$, or equivalently a weaker Coulomb interaction with the Bjerrum length of $\lambda_{B}\approx 1.0$. If we identify $C_{\infty }$ of our beads to correspond to $C_{\infty }^{\mathrm{PEO}}$ PEO monomers, the bead mass is m≈5.5×44/1.7 g/mol ≈142.5 g/mol, where 44 g/mol is the mass of a PEO monomer and $C_{\infty }^{\mathrm{PEO}}\approx 5.5$ [66].

Here we consider two different NP loadings ϕ=11.6,20.8%, for polymer chains containing charges on every third monomer, with “polymerization” degrees of N=40,200 beads per chain that have been studied before for their equilibrium properties, [42] and subject them to uniaxial elongations with different rates of ${\dot{\epsilon }}_{xx}=0.0005,\;0.001,\;0.005,\;0.01,\;0.05$.

Details of the IPNC systems studied, including NP volume fraction ϕ (%), the number of spherical NPs, $n_{\mathrm{NP}}$, polymerization degree *N*, number of chains *n*, individual NP negative charge *Q* and initial simulation box dimensions, are summarized in Table 1. A typical integration time step is dt=0.002. A representative snapshot of a fully relaxed undeformed IPNC L2${}^{•}$ is provided by Figure 1. Under these conditions, at equilibrium, the NPs prefer a body-centered cubic arrangement. See Ref. [42] for further discussions regarding the structure of the NPs at equilibrium for this model system.

### 2.3. Uniaxial Extension

We apply a uniaxial extension in the *x*-direction with constant engineering strain rates ${\dot{\epsilon }}_{xx}$ up to 800% strain. While the box size in the *x*-direction is increased deterministically as $L_{x}\left(t\right)=\left(1+{\dot{\epsilon }}_{xx}t\right)L_{x}\left(0\right)$ according to the imposed strain rate, the perpendicular box dimensions $L_{y}\left(t\right)$ and $L_{z}\left(t\right)$ are pressure-controlled, using the equilibrium pressure measured during the relaxation runs. Furthermore, at each time step, we rescale the coordinates of monomers and NPs as rigid bodies to match the dimensions of the deformed box. With this simulation setup, in the *x*-direction only the temperature is controlled via a Nosé-Hoover thermostat, while in the *y*- and *z*-direction the pressure is controlled as well via a Nosé-Hoover barostat, mimicking as closely as possible the laboratory conditions during tensile extension experiments. All systems are originally twice as long in the *y*- and *z*-directions orthogonal to the stretching *x*-direction to allow for relatively large strains at limited computational cost.

### 2.4. Quantities

The trajectories of monomers and NP centers are available at any stage of the macroscopic deformation process and allow us to evaluate the following quantities in the course of time:nonequilibrium contribution to the stress tensor, σ: its potential and kinetic contributions are calculated via the tensorial virial theorem, using coordinates and momenta.number of temporary crosslinks, $X_{c}$: between a single polymer and all NPs, averaged over all polymers. Such a temporary crosslink is formed when one (or more) monomer(s) of a chain comes into the r=1 neighborhood of an NP surface [42].entanglements per chain, $Z_{0}$: calculated using the Z1 code [67,68], in the so-called phantom limit, where NPs are removed from the system prior constructing the shortest disconnected path.NP–NP radial pair correlation function, g(r): calculated from the center coordinates of the NPs, $g\left(r\right)∼\langle \delta (r-|r_{i}-r_{j}|)\rangle /r^{2}$, where now $r_{i}$ denotes the position of an NP center, and the average is taken over all pairs of NPs.non-affine NP displacement, δ: quantifies the extent of non-affine motion of NPs at given strain. If $x_{\mu }\left(t\right)$ denotes the μth component of the time-dependent position vector of an NP center and $L_{\mu }\left(t\right)$ the box size in μ-direction, then δ is the mean length of the vectors (one for each NP), whose components are given by $x_{\mu }\left(t+\Delta t\right)-L_{\mu }\left(t+\Delta t\right)x_{\mu }\left(t\right)/L_{\mu }\left(t\right)\approx \left({\dot{x}}_{\mu }\left(t\right)-{\dot{L}}_{\mu }\left(t\right)x\left(t\right)/L_{\mu }\left(t\right)\right]\Delta t$, where Δt is the time between snapshots. We take 500 snapshots during each uniaxial extension experiment up to strain ${\dot{\varepsilon }}_{xx}t=800\%$, hence $\Delta t={\left(62.5\;{\dot{\epsilon }}_{xx}\right)}^{-1}$.mean pore radius: calculated from the pore radius histogram h(r), which offers the probability that the largest sphere that can be inserted into the polymer matrix without any overlap with existing NPs, and which contains point p and has radius *r*, provided that the insertion point p is chosen randomly from the points residing in the space available to the polymers. See for example, Ref. [7] for more details on the pore size definition.disorder parameter *S*: defined, based on the NP–NP pair correlation function, in Equation (Equation 5).radius of gyration tensor, $R_{g}$: defined as the dyadic product $\langle \left(r_{i}-c\right)\left(r_{i}-c\right)\rangle$, where $r_{i}$ denotes the position of a bead, and c the center of mass of the corresponding polymer [69,70]; the average is taken over all beads and all polymers and depends on time and strain.segment orientation tensor 〈uu〉: obtained with |u|=1 from all normalized bond vectors u along the polymer chains, as a function of strain.

Snapshots have been visualized using the visual molecular dynamics (VMD) software [71]. To help the reader follow the discussion and correlate measurements for the various quantities shown in the main text or Appendix A, we have listed the systems, regimes and content captured by all figures in Table 2.

While all quantities will be reported in dimensionless LJ units, they can be converted to dimensional numbers using the reference quantities, for which crude estimates were given above for PEO. Specifically, if a resulting quantity has a physical dimension of kg${}^{\alpha }$m${}^{\beta }$s${}^{\gamma }$C${}^{\delta }$, the dimensionless result has to be multiplied by $m^{\alpha +\gamma /2}\epsilon^{-\gamma /2}\sigma^{\beta +\gamma }{\left(e/q\right)}^{\delta }$, using m=142.5 g/mol, $\epsilon =266\;k_{B}$K, σ=1 nm, and q=7.92. For example, stress has units of Pa, that is, α=1, β=−1, γ=−2 and δ=0, implying a dimensionless stress of 1 to correspond to 3.67 MPa. Similarly, a dimensionless rate has to be multiplied with 125/ns. Fortunately, it is known that these basic considerations do not tell the whole story about the dimensional analogues [70]. The strain rates applied in the present study are unrealistically large based on this calculation, but it is still possible to reach a linear regime.

## 3. Results and Discussion

### 3.1. Stress–Strain Behavior


We begin by examining the mechanical stress–strain behavior of IPNCs. In particular, in Figure 2a,b, we show the stress–strain curves of both entangled and “unentangled” (less than a single entanglement per chain) IPNCs L1${}^{•}$ and S1${}^{•}$, for ϕ=11.6%, $\epsilon_{r}=24$ and various strain rates. In Figure 2a, it can be seen that for L1${}^{•}$ the strain rate increases the overall toughness, that is, area under the stress–strain curve. For strain rates larger than 0.001, both the stiffness and toughness of IPNCs increase with increasing strain rate. This behavior is in agreement with experiments in ionic PU-nanosilicas composites [1]. A similar behavior is observed for unentangled (N=40) IPNCs S1${}^{•}$ (Figure 2b); however, in such systems a strain of failure is found. This scenario is exemplarily visualized for S1${}^{•}$ in the case of a relatively large strain rate in Figure 3. The corresponding stress–strain curve is shown in Figure 2b.

For the entangled IPNCs L1${}^{•}$, a strain hardening regime beyond the yield point is observed [20], that is absent in the unentangled IPNCs S1${}^{•}$. Furthermore, the insets in Figure 2a,b indicate that at small strains ($\epsilon_{xx}\leq 10\%$) and at low strain rates (${\dot{\epsilon }}_{xx}\leq 0.001$) a linear regime for both L1${}^{•}$ and S1${}^{•}$ systems emanates where the entangled systems show a larger Young’s modulus (initial slope of the stress–strain curve) than the unentangled ones. Both systems exhibit a stress overshoot at the beginning of extension for the largest strain rates (${\dot{\epsilon }}_{xx}=0.05$) consistent with the experimental observations for polymer melts.

Moreover, in Figure 4a(L systems), b(S systems), we examine the effects of the dielectric constant (or Bjerrum length) [42] and NP volume fraction on the stress–strain behavior for a selected strain rate ${\dot{\epsilon }}_{xx}=0.01$ (cyan curves in Figure 2). As can be seen in both polymer cases, the toughness of the IPNC increases substantially for a higher Bjerrum length (smaller dielectric constant), which means a stronger electrostatic strength (in short, toughness is larger in • systems than ∘ systems). However, the stiffness of the IPNCs (the initial slope of the stress–strain curves) does not seem to be influenced by the electrostatic strength. The NP volume fraction influences the stiffness and toughness of the IPNCs. It can be clearly seen that the addition of ionic NPs into the polymer matrix leads to a substantial decrease of toughness (in short, toughness 1>2). Additional results for the stress–strain behavior of IPNCs L2${}^{•}$ and S2${}^{•}$ at a larger NP volume fraction of ϕ=20.8%, for various strain rates analogous to those investigated for L1${}^{•}$ and S1${}^{•}$ (Figure 2), are provided in Appendix A. It is worth noting that the modeled stress–strain behavior in INPCs, both stiffness and toughness, is enhanced compared with the modeled stress–strain curve of traditional (nonionic) nanocomposites [24,25].

### 3.2. NP Structure during Uniaxial Extension


In this section, we focus on the NP structure during the deformation process of the IPNCs. In Figure 4a we can see the pronounced stress overshoot. This phenomenon is accompanied by a transition between affine and non-affine motion of the NP subsystem, quantified in terms of the non-affine displacement δ (Appendix A) introduced in Section 2.4; δ begins to rise suddenly at the inflection point of the stress, that is, the stress increase triggers a non-affine motion of NPs, and vice versa, the non-affine motion tends to help relax the stress. In general, the degree of non-affinity is comparable at very low and very high strains and reaches its maximum at intermediate strains.

To quantify the structural transition further, we show in Figure 5b the NP–NP radial pair correlation function, g(r), for the L2${}^{•}$ system at an intermediate strain rate ${\dot{\epsilon }}_{xx}=0.005$, for various selected strains marked by bullets in the stress–strain curve shown in Figure 5a. Compared with the non-affine displacement, which characterizes changes that may leave g(r) unchanged, g(r) can be considered as a more classical measure of order than δ, while it should be kept in mind that it is most useful for isotropic systems. Snapshots that correspond to the states marked by bullets up to a strain of 200% are provided in Figure 6. The first of these snapshots coincides with the equilibrium snapshot shown in Figure 1. According to these results, the strain has a distinct effect on the structure of NPs. For low strains, up to the stress peak at a strain of about 70%, there is more than one peak (and characteristic distance) in the pair correlation function signaling a pronounced NP ordering. Actually, while the system exhibits a near-crystalline body-centered cubic order in equilibrium (Figure 1), it transits through a cascade of long range ordered structures including a hexagonal close packed (hcp) structure, during which the stress rises. The transition between body-centered cubic and hexagonal close-packed can be easily seen in the first two (blue and green) curves of Figure 5b. Up to distances of twice the minimum NP–NP distance, the body-centered cubic can be recognized by a large first peak, followed by two peaks of comparable magnitude, while the hexagonal close packed as opposed to face-centered cubic has a peak at about r=21, just before a larger peak emanates that would also occur for the face-centered cubic structure. Disorder has started to set in at the yield point (marked by a red bullet). For higher strains, there is only one broad peak remaining in the pair correlation function representing a random packing of the ionic NPs in the polymer matrix, also confirmed by the snapshots of Figure 6. To quantify the amount of disorder reflected by the shape of the pair correlation function, not only for selected strains but also over the whole range of strains, we introduce a simple entropy-like disorder parameter based on g(r),
(5)$$S=-\frac{\int g(r)\ln[g(r)]\;dr}{\int g(r)\;dr},$$
where the integral extends over regions in *r*, where g(r) does not vanish. The quantity *S* gives information about the degree of disorder; *S* vanishes for an ideal gas, and gets progressively negative with increasing order. The behavior of *S* for L2${}^{•}$ as a function of strain is shown in Figure 5c. At low strains, the system remains long-range ordered, while the structure reorganizes, starting out with a body-centered cubic NP structure. Beyond strains of about 50%, *S* sharply increases until it reaches a maximum at about $\varepsilon_{xx}\approx 80\%$. Just at the maximum the long-range order vanishes completely and the unordered structure remains up to the largest strains probed here, while there is a moderate tendency for an increase of order with increasing strain, that is not visible in g(r), but quantified by *S* (Figure 5c). Results for the L2${}^{•}$ at different strain rates are provided in Section 3.7 and Appendix A. Analogous results for the more NP-dilute L1${}^{•}$ are available in Appendix A.

### 3.3. Polymer Conformations

Beyond the NP structure, it is of interest to investigate the polymer conformations, which affect the polymer–polymer entanglements [59,72,73] during the uniaxial extension. Here, we elaborate on the polymer conformations by investigating components of gyration tensor, $R_{g}$, for different numbers of monomers and NP volume fractions, during the uniaxial extension process. It is worth recalling that all ionic polymers exhibit the same charge density. Thus, the intramolecular electrostatic interaction between monomers has the same effect on polymer dimensions and entanglements in all systems studied. Changes of polymer conformations, entanglements and ionic crosslinks may therefore arise from the electrostatic attraction between the NPs and ionic polymers, which depends on the NP volume fraction and Bjerrum length, as the charge per NP is decreasing by increasing NP volume fraction.

First, the size of polymers in the absence of deformation is only weakly affected by NP volume fraction (Figure 7) or strength of the electrostatic interactions (data not shown). In particular, components of the gyration tensor as a function of strain for all weakly electrostatic systems L${}^{\circ }$ and S${}^{\circ }$ at a rate of ${\dot{\epsilon }}_{xx}=0.01$, for which stress–strain curves were presented in Figure 4, are shown in Figure 7. We observe a characteristic increase of the parallel component of the gyration tensor, $R_{gxx}$, to the expense of a decrease on its perpendicular component $(R_{gyy}+R_{gzz})/2$ with respect to the extension direction. The behavior of these two components as a function of strain is qualitatively similar to the ideal, uniaxially extended, polymer coil, which is limited in its extension by the polymerization degree ($R_{gxx}\leq {\left(N-1\right)}^{2}/12$ because the bond length is approximately unity). The parallel and perpendicular components of $R_{g}$, for entangled chains, are slightly enhanced for the lower NP volume fractions during the extension (L2${}^{\circ }$) since, in that case, the NPs carry a higher total charge, which leads to a stronger electrostatic attraction between polymer and NPs. Additional data for gyration tensor components versus strain at several rates for the L1${}^{•}$ and S1${}^{•}$ systems are provided in Appendix A, respectively. The stress–strain relationships can be correlated with the conformations of the polymers to investigate the stress-optic behavior; this is done in Appendix A.

### 3.4. Polymer–Polymer Entanglements

In this section, we focus on the polymer entanglements during the extension for all four L systems, using the method described in Section 2.4. It is an often quoted statement that entanglements contribute to the stress–strain behavior of the IPNC, and specifically to the toughness and strain hardening regime of the material. We find that the number of entanglements per chain is generally reduced with increasing strain rate, beyond a strain rate of 0.001, as it can be seen in Figure 8a for the representative L1${}^{•}$ system. It is worth noting that the number of entanglements $Z_{0}$ is first rising for strains below 300%, and dropping afterwards. This is because chains tend to first stretch during deformation and increase their number of polymer–polymer contacts, before they lose entanglements due to alignment. While there is a clear reduction of $Z_{0}$ with increasing NP volume fraction and decreasing NP–NP distance (Figure 8), the Bjerrum length does not seem to have any effect on $Z_{0}$ for the high NP volume fraction, L2${}^{\circ }$ and L2${}^{•}$ systems. At the low NP fraction, however, an increase of Bjerrum length is accompanied by an increase of entanglements at sufficiently low strains (Figure 8a), in short: entanglements •≥∘ and 1>2 and their amount decreases with increasing rate. It is worth noting that the number of polymer entanglements is higher for lower NP volume fraction IPNCs, due to the stronger electrostatic attraction between polymers and NPs for that case. This higher number of entanglements at a lower NP volume fraction is accompanied by strain hardening and a higher toughness of the IPNC as is depicted in Figure 4a for all L systems.

However, although there are regimes with the same number of polymer entanglements in the plasticity region, the corresponding toughness and strain hardening values of the IPNCs are different. Such results clearly denote the correlation of electrostatic strength (attraction) with the toughness and strain hardening behavior of the IPNC. In short: while toughness and entanglements are both •≥∘ and 1>2, toughness increases with rate, while entanglements decrease with rate. Thus, a direct correlation between entanglements and toughness (or strain hardening regime) can be ruled out.

An increase of Bjerrum length tends to reduce the number of entanglements for the systems at low NP volume fraction (ϕ≈11.6%); however, it does not alter the entanglements at larger volume fractions (ϕ≈20.8%), especially in light of the results for the S systems (Appendix A). In essence, stronger electrostatic interactions (attraction) seem to enhance the effect of NP content on the number of entanglements, for all systems.

### 3.5. Polymer-NP Temporary Crosslinks

Next, we focus on the ionic temporary crosslinks (Section 2.4) created between NPs and polymers during extension. We have shown in our previous study by equilibrium MD [8,42], that there are ionic temporary crosslinks formed, depending on the electrostatic strength and charge sequence of the ionic polymers. It is shown in Figure 9a for the L2${}^{•}$ system that low strain rates have a very subtle effect on the number of crosslinks $X_{c}$. As the strain rate increases to 0.05, there is a substantial decrease of crosslinks with increasing strain. For all rates, the number of crosslinks increases with strain. It can be seen in Figure 9b that Bjerrum length (different dielectric constant) has a distinct effect on the number of ionic crosslinks, especially at 20.8% NP volume fraction (systems L2${}^{•}$ and L2${}^{\circ }$); in particular, a lower Bjerrum length increases the number of crosslinks at any strain. In short: crosslinks 1<2 and •≤∘ and their amount increases with strain. This can be explained by the fact that at a lower Bjerrum length ($\epsilon_{r}=50$), due to the weaker electrostatics strength, NPs and polymers have a higher mobility thus ionic crosslinks can be formed more easily. In light of results shown in Figure 4a for all L systems, it can be seen that at a lower Bjerrum length, the toughness of the IPNC reduces, which means that the mobility of NPs is not the main mechanism for driving the toughness behavior of the IPNC but rather the strength of ionic crosslinks formed. This result comes into agreement with the simulation work of Kutvonen et al. [18] in which the minimization of NPs mobility was reported to contribute to the toughness of a polymer nanocomposite but it disputes what is claimed in the experimental work by Odent et al. [1] Thus, the mechanism that drives the toughness of the IPNC is the electrostatic strength of the ionic crosslinks formed between polymers and NPs. Obviously, for lower NP volume fractions (systems L1${}^{•}$ and L1${}^{\circ }$), there are fewer NP–polymer crosslinks formed due to the smaller number of available NPs. However, as can be seen in Figure 4a, a better toughness behavior and a greater strain hardening for such lower NP loading IPNCs appears. Such behavior can also be explained by the strength of ionic crosslinks formed, since at lower NP volume fractions, NPs contain a higher total charge, thus a stronger electrostatic attraction between polymers and NPs exists.

The number of ionic crosslinks are generally reduced in the unentangled S systems compared to the entangled L systems. Still, we present the full analysis of the low-molecular weight systems in Appendix A for S2${}^{•}$, where we have varied the strain rate, and in Appendix A at a selected rate for all four S systems, for a direct comparison with results discussed in the current section. While there are qualitative similarities between the L and S systems, the amount of ionic crosslinks increases less dramatically with strain for the S systems. For the highest strain rate probed, we even find $X_{c}$ to go through a minimum at a strain of about 400%, while the L systems have only a very moderate increase instead of a minimum. This effect is due to the ability of polymer chains to stretch by a factor ∼$\sqrt{N}$, the ratio between fully extended and coiled equilibrium state, before they detach.

### 3.6. Pore Size Analysis


To shed some light on the remarkable structural, anisotropic changes of the deformed NP subsystem, that are captured only in very crude fashion by the isotropic part of the pair correlation function, we evaluate the pore size distribution (Section 2.4) and its first moment, the mean pore radius, as a function of strain. Let us begin by inspecting the mean pore radius $r_{p}$ versus strain for the system L1${}^{•}$ for which we have already monitored the nonequilibrium stresses and gyration tensor components, subjected to different strain rates (Figure 10a). We find that the peak in the mean pore radius occurs exactly at an order–disorder transition, and its value, that is, the size of pores, tends to increase with increasing strain rate. This implies that our initially ordered systems are able to transit through intermediate, differently but still very well ordered states with larger pore sizes on average. For the remaining L systems, the pore radius is shown versus strain for the selected rate ${\dot{\varepsilon }}_{xx}=0.01$ in Figure 10b. In particular, NP volume fraction has a clear effect on $r_{p}$, denoting a larger $r_{p}$ for lower volume fractions. In addition, a higher Bjerrum length leads to a higher $r_{p}$; beyond the strain the peak of $r_{p}$ appears. This is due to the stronger electrostatic attraction that exists between polymers and NPs and keeps NPs closer in the neighborhood of polymers during the extension for a larger Bjerrum length.

This information is very different from that contained in g(r). While the first peak in g(r) denotes the preferred distance between neighboring NPs, and remaining peaks could be used to speculate about the anisotropic ordering, the mean pore size provides orthogonal information about the three-dimensional arrangement. The largest pore radius is observed for a configuration that is close to mono–modal simple cubic (SC) packing. It is well known that the pore size of an SC lattice is a factor 0.55/0.41≈1.34 larger than that of the initial body-centered (BCC) cubic lattice [74]. This ratio equals the ratio 7.4/5.5≈1.34 observed for the largest strain rate in Figure 11a.

An analogous analysis for the S systems is available in Appendix A for S1${}^{•}$ and in Appendix A for all unentangled S systems.

### 3.7. Stretching of the Entangled IPNC Systems at Low Strain Rates

By lowering the strain rate further, we begin to enter a regime where self-strengthening features of entangled IPNCs can be clearly observed. While the L2${}^{•}$ system was shown to exhibit strain-hardening behavior at a rate of ${\dot{\varepsilon }}_{xx}=0.005$, the stress–strain curves for both L2 systems, at a five times smaller rate ${\dot{\varepsilon }}_{xx}=0.001$, are provided in Figure 11a. Corresponding snapshots for the selected strains, marked by colored bullets, are provided in Figure 12. Instead of a single local minimum at a finite strain, we observe a noisy stress signal at first glance. However, the results, presented here, are obtained starting from more than ten independent initial configurations, and the stress signal shown in Figure 11a is reproducible to within a few percent. The physical mechanism behind this self-strenghening zig-zag signal is a build-up of highly ordered structures at certain strains, which renders similarity with stress–strain signals obtained for mechano-ionic switches [75,76]; In between those strains, the system reorganizes in a relatively unordered fashion, as it apparently lacks the possibility of falling into a global energetic minimum. As for the IPNCs discussed earlier in this manuscript, δ rises at the inflection point of the stress, that is, a stress increase triggers a non-affine motion and vice versa, the non-affine motion tends to relax the stress (Appendix A). Contrary to Figure 4a, we now observe more than a single overshoot. Structural reorganization produces a configuration for which the affine extension leads to a stress increase, and the process is repeated until the non-affine displacement results in a random or less structured state.

By visual inspection of Figure 12, we can observe that the various ordered phases do not exhibit the same crystal symmetry. To make this more quantitative, we have calculated the NP–NP pair correlation function g(r), depicted in Figure 11b. The alternating aspects of both quantities, snapshots and correlation function are obvious. In particular, it can be seen that the g(r) graphs (blue, red, magenta lines) with a few localized peaks for strains 0%, 150%, and 410% alternate with g(r) graphs (green, cyan, yellow lines) that exhibit a single and very broad peak at a value that is determined by the number density of NPs. The corresponding maxima of the non-affine displacement δ occur with a delay, for a reason discussed already, at the larger strains 60%, 220%, and 550% (Appendix A), independent of the strength of the electrostatic interaction.

In addition, the crystal-like g(r) informs us that, while there is a body-centered cubic arrangement of the NPs at equilibrium, it transforms into face-centered cubic structures at the finite strains ≈150% and ≈410%. For a discussion of deformation-induced transitions between lattice types we refer to Wang et al. [77]. As the snapshots clearly reveal, the lattice has been first destroyed and then rebuilt at another angle with respect to the extension direction. The stress at these ordered structures is particularly low, giving rise to the observed zig-zag transients. The complete time-dependent behavior of the pair correlation function, characterized by the disorder parameter *S*, is shown in Figure 11c. It can be seen that both the stress–strain and ordering behavior are very similar for weak and strong electrostatics (different dielectric constants), while the stresses are larger for the L2${}^{•}$ system.

## 4. Conclusions

Ionic polymer nanocomposites (IPNCs) under high uniaxial extension were studied by means of extensive nonequilibrium molecular dynamics simulations. The effects of the nanoparticle (NP) loading, polymer matrix molecular weight, electrostatic strength, and engineering strain rate on the mechanical and structural properties of the IPNCs under uniaxial extension were investigated.

The simulation results revealed that IPNCs with an entangled polymer matrix can withstand larger strains, have greater toughness and show strain hardening relative to the IPNCs with an unentangled polymer matrix under the same loading conditions. In accordance with recent experiments, a larger strain rate resulted in a larger toughness for all systems studied here. The results further suggest that the mechanical properties of IPNCs are not governed solely by the number of polymer–polymer entanglements and/or NP mobility, but also more strongly by the temporary ionic polymer–NP crosslinks, the strength of ionic interactions, and the NP loading.

At relatively low strain rates, IPNCs with an entangled polymer matrix and large NP loading showed self-strengthening properties similar to those of mechano-ionic switches. The NP lattice structure indicated “healing” after being destroyed (during extension) in a repeated manner, resulting in a zigzag-type stress response up to very large strains (800%).

The present study reveals detailed microscopic insight into the complex interplay between structural changes accompanying the uniaxial mechanical deformation behavior of IPNCs, the roles played by the ionic interactions, the evolution of the entanglement network, and temporary ionic crosslinks between NPs and polymers. The structures have been characterized in great detail, from snapshots to pair correlation functions and pore size distributions; only part of this information is experimentally accessible at present. Further studies may focus on the linear viscoelastic and shearing behavior [78] of IPNCs in comparison with their neutral counterparts, as there are additional rotational components that need to be explored to complete our understanding of the nonequilibrium mechanical behaviors of IPNCs.

## Figures and Tables

**Figure 1 polymers-13-04001-f001:**
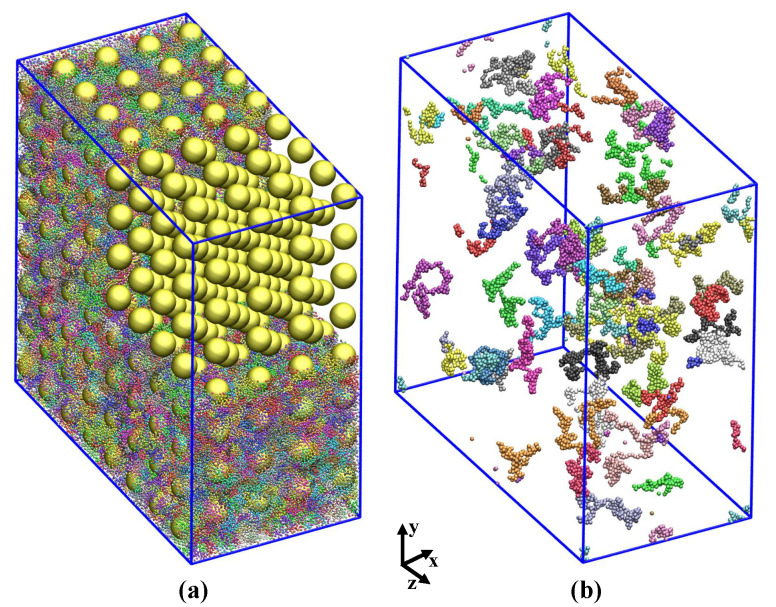
(**a**) Simulation snapshot of system L2${}^{•}$, a fully relaxed undeformed IPNC system at NP volume fraction ϕ=20.8%, dielectric constant $\varepsilon_{r}=24$, and monodisperse chains with N=200 beads each. The small, colorful spheres depict the connected polymer beads, and the large golden ones stand for the NPs. Note that the surface of each NP is covered homogeneously with 720 smaller NP surface beads that are not shown here for the sake of clarity. Polymers in the top corner are removed so that the NP (almost perfect lattice) structure is visible. The system contains n=2304 polymer chains, nN=460800 polymer beads and 512 NPs in total (Table 1). (**b**) The same system in which only 50 randomly selected chains, each with a distinct color, is shown while the rest of the chains as well as the NPs are invisible.

**Figure 2 polymers-13-04001-f002:**
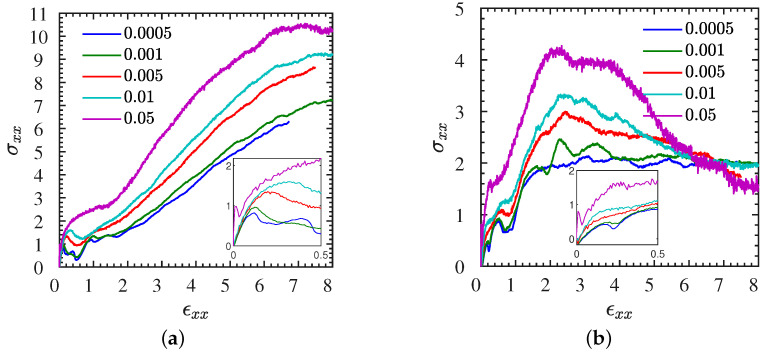
Rate-dependence of stress–strain curves for (**a**) entangled L1${}^{•}$, and (**b**) unentangled S1${}^{•}$ at an NP loading of ϕ=11.6%, and $\varepsilon_{r}=24$. Different rates are marked by different colors, from ${\dot{\varepsilon }}_{xx}=0.0005$ (blue) to ${\dot{\varepsilon }}_{xx}=0.05$ (purple). Insets provide zooms into the regions of moderate strains below 50%.

**Figure 3 polymers-13-04001-f003:**
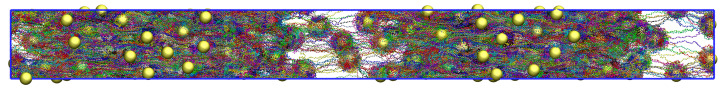
Snapshot taken from the S1${}^{•}$ system. Breaking of the unentangled IPNC system under large uniaxial extension strain $\epsilon_{xx}=750\%$ at (fast) engineering extension rate of ${\dot{\epsilon }}_{xx}=0.05$, c.f. purple line in Figure 2b. The simulation snapshot corresponds to a system with N=40 chains at an NP loading of ϕ=11.6%, and $\varepsilon_{r}=24$.

**Figure 4 polymers-13-04001-f004:**
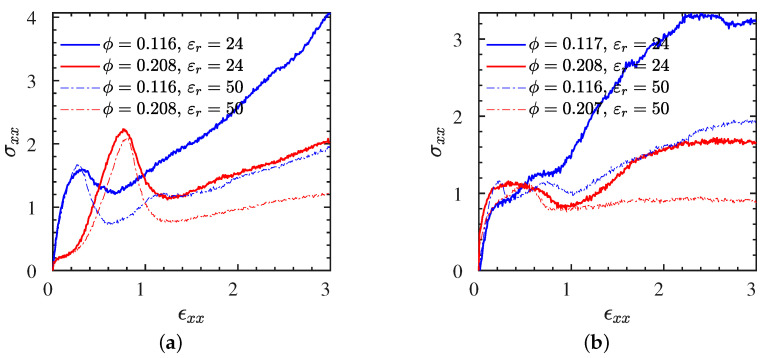
All eight L and S systems at identical engineering strain rate. Stress–strain curves for (**a**) L (N=200), and (**b**) S (N=40) systems at different NP loadings and dielectric constants subjected to a stretching rate of ${\dot{\epsilon }}_{xx}=0.01$.

**Figure 5 polymers-13-04001-f005:**
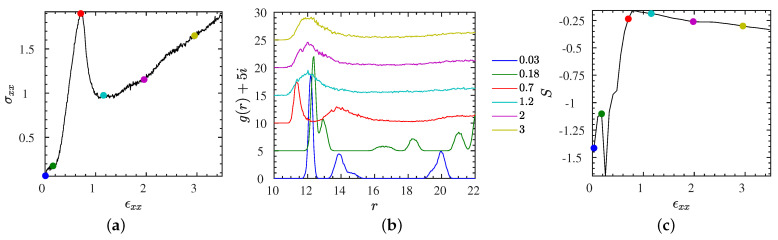
System L2${}^{•}$ subjected to strain rate ${\dot{\epsilon }}_{xx}=0.005$ at different stretching strains. (**a**) stress–strain curve. (**b**) NP–NP pair correlation function at the strains marked by colorful bullets in (**a**). The curves are shifted upward by their index i=0,...,5 where, e.g., i=0 for the blue line (0.03 strain) and i=5 for the yellow line (3 strain). (**c**) Corresponding information measure *S* obtained from g(r) shown in (**b**) via Equation (Equation 5). Snapshots for the marked states are provided in Figure 6.

**Figure 6 polymers-13-04001-f006:**
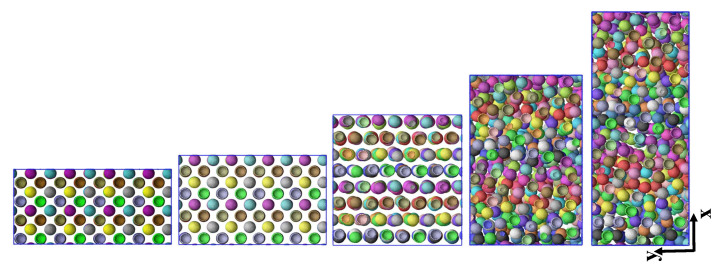
Snapshots (projections onto the *x*-*y*-plane) of the three-dimensional L2${}^{•}$ system with N=200, $\varepsilon_{r}=24$, ϕ=20.8%, subjected to uniaxial extension in *x*-direction at strain rate ${\dot{\epsilon }}_{xx}=0.005$, captured at different stretching strains of $\epsilon_{xx}=0$, 18%, 70%, 120%, and 200% from top to bottom. These states are marked by colored bullets in Figure 5a,c. The NPs remain unstructured at strains larger than $\epsilon_{xx}\geq 200\%$. Polymers are not shown here for the sake of clarity. The corresponding NP–NP pair correlation functions at these strains are given in Figure 5b.

**Figure 7 polymers-13-04001-f007:**
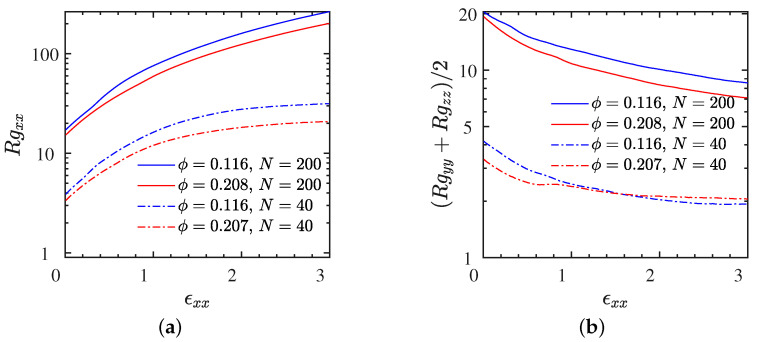
Polymer structure during stretching for different NP loadings and chain lengths for all weakly electrostatic L${}^{\circ }$ and S${}^{\circ }$ systems ($\varepsilon_{r}=50$). Different elements of gyration tensor, (**a**) parallel- and (**b**) perpendicular to the direction of strain. Data correspond to the strain rate of ${\dot{\epsilon }}_{xx}=0.01$. A similar trend was observed for strongly electrostatic systems L${}^{•}$ and S${}^{•}$ but with a slight increase (less than 10%) in gyration tensor elements (results not shown here). Additional data highlighting the effect of strain rate provided in Appendix A.

**Figure 8 polymers-13-04001-f008:**
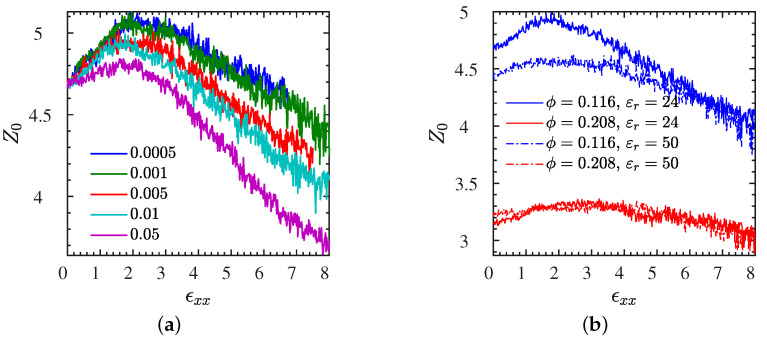
Entanglement data for all L systems. Number of entanglements per chain, $Z_{0}$, in the phantom NP limit during stretching for the L systems containing N=200 chains. Dependence of $Z_{0}$ on stretching rate ${\dot{\epsilon }}_{xx}$ (**a**) for L1${}^{•}$. (**b**) $Z_{0}$ dependence on ϕ and $\varepsilon_{r}$ measured at a stretching rate of ${\dot{\epsilon }}_{xx}=0.01$. The corresponding stress–strain curves are given in Figure 2a and Figure 4b respectively. Additional data for the S systems available in Appendix A.

**Figure 9 polymers-13-04001-f009:**
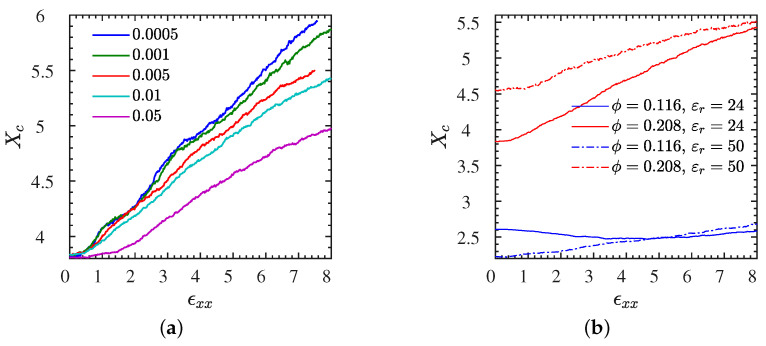
Crosslink data for all L systems. (**a**) Polymer–NP temporary crosslinks per chain $X_{c}$ as a function of elongational strain $\epsilon_{xx}$ for systems with dielectric constant and NP loading of $\varepsilon_{r}=24$ and ϕ=20.8% respectively, measured at different strain rates. (**b**) $X_{c}$ measured at various ϕ and $\varepsilon_{r}$ for samples subjected to a strain rate of ${\dot{\epsilon }}_{xx}=0.01$. All data shown here correspond to entangled systems with N=200 chain length. Data for the short chain S systems available in Appendix A.

**Figure 10 polymers-13-04001-f010:**
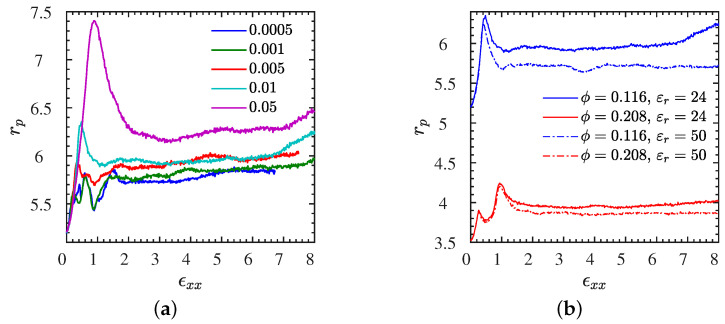
Pore radius data for all L systems. (**a**) Dependence of the mean pore radius $r_{p}$ on stretching rate ${\dot{\epsilon }}_{xx}$ for the L1${}^{•}$ system at various rates. (**b**) $r_{p}$ dependence on ϕ and $\varepsilon_{r}$ measured at a stretching rate of ${\dot{\epsilon }}_{xx}=0.01$. The corresponding stress–strain curves are given in Figure 2a and Figure 4a respectively. Analogous analysis for the S systems in Appendix A for S1${}^{•}$ and in Appendix A for all S systems.

**Figure 11 polymers-13-04001-f011:**
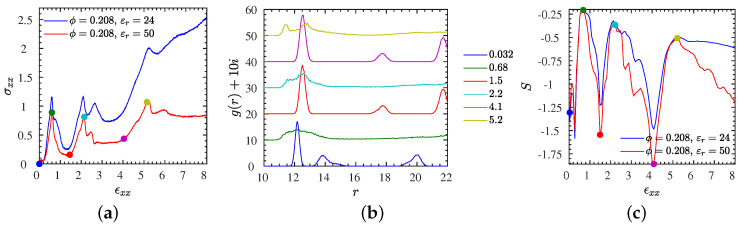
Structural transition data for the entangled L2${}^{•}$ and L2${}^{\circ }$ systems at an identical and relatively small strain rate ${\dot{\epsilon }}_{xx}=0.001$. (**a**) Stress–strain curves for L2${}^{•}$ (blue) and L2${}^{\circ }$ (red lines), (**b**) NP–NP pair correlation function calculated at different stretching strains for the L2${}^{\circ }$ system. The curves are shifted upward by their index i=0,...,5 where, e.g., i=0 for the blue (bottom) line and i=5 for the yellow (top) line. The line colors in (**c**) correspond to the observations (colorful symbols) made in (**a**,**c**) at the same strain $\epsilon_{xx}$ values. (**c**) Disorder parameter *S* versus strain for both L2${}^{•}$ and L2${}^{\circ }$ systems. A similar analysis for the L2${}^{•}$ system at a larger rate was provided by Figure 5.

**Figure 12 polymers-13-04001-f012:**
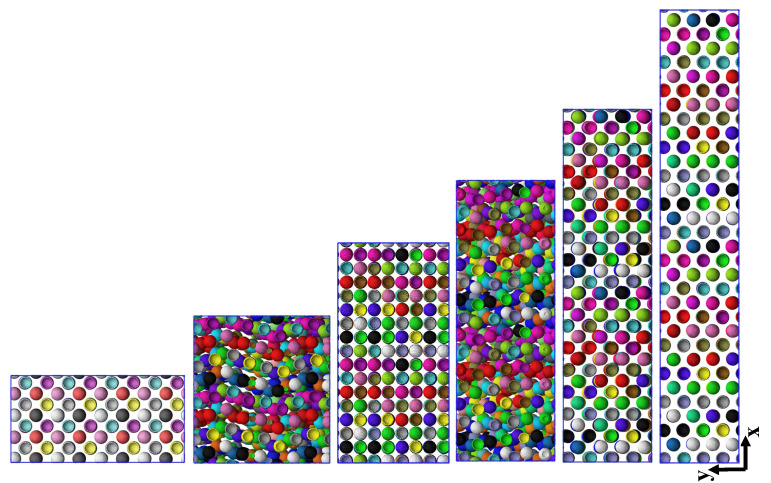
L2${}^{\circ }$ system at strain rate ${\dot{\epsilon }}_{xx}=0.001$, captured at different stretching strains of $\epsilon_{xx}=0$, 68%, 150%, 220%, 300%, and 410% from top to bottom respectively. Polymers are not shown here for the sake of clarity. The corresponding NP–NP pair correlation functions, stresses, and disorder parameters at these strains are given in Figure 11. All the coordinates here are projected onto the xy plane. We have chosen only to show L2${}^{\circ }$ here; the snapshots are very comparable for L2${}^{•}$, as Figure 11c also implies.

**Table 1 polymers-13-04001-t001:** IPNCs studied in this work. Polymerization degree *N*, NP volume fraction ϕ, relative dielectric constant $\varepsilon_{r}$ [43], number $n_{\mathrm{NP}}$ of spherical NPs of identical radius $r_{\mathrm{NP}}$, number of polymer chains *n*, individual NP negative charge *Q*, and initial simulation box dimensions $L_{x}\times L_{y}\times L_{z}$. Every 3rd polymer bead is periodically charged along the polymer backbone (including chain ends). The number of charges per polymer is $n_{\mathrm{NP}}Q/n$ in each system. Each of the eight systems, for which we introduce system labels in the first column, was subjected to uniaxial extension using five different strain rates: ${\dot{\epsilon }}_{xx}=0.0005$, 0.001, 0.005, 0.01, and 0.05. The label L1${}^{•}$ denotes a system containing long (L) chains at low NP volume fraction (1) in the presence of strong electrostatic interactions (•), while S2${}^{\circ }$ denotes a system containing short (S) chains at high NP volume fraction (2) in the presence of weak interactions (∘), and so on. If one of the three identifiers is missing as in S2, we refer to short chain, NP-dense systems with both weak and strong interactions.

Label	*N*	ϕ	$e_{r}$	$n_{\mathrm{NP}}$	*n*	Q/q	$L_{x}\times L_{y}\times L_{z}$
L1${}^{•}$	200	11.6%	24	256	2304	603	54.2 × 108.3 × 108.3
L1${}^{\circ }$	200	11.6%	50	256	2304	603	54.2 × 108.3 × 108.3
L2${}^{•}$	200	20.8%	24	512	2304	306	56.2 × 112.4 × 112.4
L2${}^{\circ }$	200	20.8%	50	512	2304	306	56.2 × 112.4 × 112.4
S1${}^{•}$	40	11.7%	24	192	8640	630	49.3 × 98.5 × 98.5
S1${}^{\circ }$	40	11.6%	50	192	8640	630	49.3 × 98.5 × 98.5
S2${}^{•}$	40	20.8%	24	384	8640	315	51.1 × 102.3 × 102.3
S2${}^{\circ }$	40	20.7%	50	384	8640	315	51.1 × 102.3 × 102.3

**Table 2 polymers-13-04001-t002:** Guide through results shown by figures in both main text and Appendix A. System labels have been introduced in Table 1, strain rates ${\dot{\varepsilon }}_{xx}$, strain $\varepsilon_{xx}$ and quantities are mentioned here. Parts of labels have been left out if the corresponding figure shows results for all labels that contain the pattern such as S2; it stands for S2${}^{•}$ and S2${}^{\circ }$. Additional data available from authors upon request.

Figure	Label	${\dot{\epsilon }}_{\mathrm{xx}}$	$\epsilon_{\mathrm{xx}}$	Content	Figure	Label	${\dot{\epsilon }}_{\mathrm{xx}}$	$\epsilon_{\mathrm{xx}}$	Content
Figure 1	L2${}^{•}$	0	0	snapshot	Appendix A	L2${}^{•}$, S2${}^{•}$	varied	0–8	stress–strain
Figure 2a,b	L1${}^{•}$, S1${}^{•}$	varied	0–8	stress–strain	Appendix A	L2${}^{•}$	varied	0–5	correlation g(r)
Figure 3	S1${}^{•}$	0.05	7.5	snapshots	Appendix A	L1${}^{•}$	varied	0–3	correlation g(r)
Figure 4a,b	L, S	0.01	0–3	stress–strain	Appendix A	L1${}^{•}$	varied	0–8	gyration $R_{g}$
Figure 5	L2${}^{•}$	0.005	0–3	stress–strain, g(r), *S*	Appendix A	S1${}^{•}$	varied	0–8	gyration $R_{g}$
Figure 6	L2${}^{•}$	0.005	0–2	snapshots	Appendix A	L, S	0.01	0–8	stress-optic
Figure 7	L${}^{\circ }$, S${}^{\circ }$	0.01	0–3	gyration $R_{g}$	Appendix A	S1${}^{•}$	varied	0–8	entanglements $Z_{0}$
Figure 8>a	L1${}^{•}$	varied	0–8	entanglements $Z_{0}$	Appendix A	S	0.01	0–8	entanglements $Z_{0}$
Figure 8b	L	0.01	0–8	entanglements $Z_{0}$	Appendix A	S2${}^{•}$	varied	0–8	crosslinks $X_{c}$
Figure 9a	L2${}^{•}$	varied	0–8	crosslinks $X_{c}$	Appendix A	S	0.01	0–8	crosslinks $X_{c}$
Figure 9b	L	0.01	0–8	crosslinks $X_{c}$	Appendix A	S1${}^{•}$	varied	0–8	pore $r_{p}$
Figure 10a	L1${}^{•}$	varied	0–8	pore $r_{p}$	Appendix A	S	0.01	0–8	pore $r_{p}$
Figure 10b	L	0.01	0–8	pore $r_{p}$	Appendix A	L2	0.001	0–8	$R_{g}$, $Z_{0}$, $r_{p}$
Figure 11	L2	0.001	0–8	stress–strain, g(r), *S*	Appendix A	L, S	0.01	0–3	non-affine δ
Figure 12	L2${}^{\circ }$	0.001	0–5	snapshots	Appendix A	L2	0.001	0–8	non-affine δ

## Data Availability

Data is available online as part of the SNF-200021L-185052-repository at https://doi.org/10.3929/ethz-b-000502732 (accessed on 18 November 2021).

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
