# Peer review of "Ionic Polymer Nanocomposites Subjected to Uniaxial Extension: A Nonequilibrium Molecular Dynamics Study"

_polymers, 2021, doi:10.3390/polym13224001_

Round 1

Reviewer 1 Report

The authors studied ionic polymer nanocomposites under uniaxial extension by using coarse-grained molecular dynamics. The effects of entanglement of polymer chains, the fraction of nanoparticle, dielectric constant (Bjerrum length) on the elastic properties and structures are reported. These results and the detail of analysis provide interesting aspects of unique properties of ionic nanocomposites comparing well studied nonionic ones. So, I recommend publishing this article on Polymers. However, I have one concern about the computational model and corresponding results,

The amount of surface charge of nanoparticles depends on the volume fraction to maintain charge neutrality of the system, and the electrostatic interaction of each particle should be different on different volume fraction. So, the results of different volume fraction in Figure 4 and 8 for example contain the effect of volume fraction and electrostatic interaction. The authors should clear the effects of the electrostatic interaction and the volume fraction.

Author Response

We thank this referee for the comment. We agree that we need to keep the whole nanocomposite system neutral, in order to have a charge balance between ionic polymers and nanoparticles. In our ionic nanocomposite, we do not include counterions, to compensate for charge, as they are absent from the experimental system (Reference 1) as well. 
If the total charge of NPs were chosen to be the same for different NP loadings then the polymers charge density would be different (for various NP loadings) in order to keep neutrality (charge balance). That would result in significantly different polymer conformations and dimensions, due to the intramolecular electrostatic interaction of monomers, and thus to different polymer-entanglements and ionic crosslinks. And vice versa, if the polymer charge density is kept constant, the NP charge depends on the NP volume fraction. We have chosen the latter approach, as it is the one encountered in the experiment, and 
because this way we exclude any effect of polymer charge density on polymer entanglements and ionic crosslinks. We can, thus, clearly see and discuss the effect of NP loading and Bjerrum length (electrostatic strength) on stress-strain behavior, entanglements and crosslinks in Figures 4a, 7a, 8b, 9b.

Therefore we have added into the section 2.1:

" to satisfy the charge balance between ionic polymers and nanoparticles (Ref. 1). For both NP volume fractions investigated, the ionic polymers keep their chemistry, i.e., their unaltered charge density, as in the experiment (Ref. 1). We do not include counterions, to compensate for the charge, since they do not appear in the experimental nanocomposite system (Ref. 1) as well. This implies that the NP charge depends on the NP volume fraction and that the effects of charge and NP volume fraction cannot be explored independently, without 
altering the polymer dimensions."

and we also added in Section 3.3, 

"It is worth recalling that all ionic polymers exhibit the same charge density. Thus, the intramolecular electrostatic interaction between monomers has the same effect on polymer dimensions and entanglements in all systems studied.
Changes of polymer conformations, entanglements and ionic crosslinks may therefore arise from the electrostatic attraction between the NPs and ionic polymers, which depends on the NP volume fraction and Bjerrum length, as the charge per NP is decreasing by increasing NP volume fraction"

Reviewer 2 Report

This manuscript is an representation of remaining evaluation of work [44]. It can be proceeded further for publication after incorporating the following query.

  • The sentences starting with “line 85-89” and “line 67-71” has the length of a paragraph. Better to rephrase it.

Author Response

We agree with the Referee and we have shortened the sentences in the mentioned paragraphs. All changes have been highlighted in the revised manuscript.